# Impact of COVID-19 Infection on Children and Adolescents after Liver Transplantation in a Latin American Reference Center

**DOI:** 10.3390/microorganisms10051030

**Published:** 2022-05-15

**Authors:** Aline F. Freitas, Renata P. S. Pugliese, Flavia Feier, Irene K. Miura, Vera Lúcia B. Danesi, Eliene N. Oliveira, Adriana P. M. Hirschfeld, Cristian B. V. Borges, Juliana V. Lobato, Gilda Porta, João Seda-Neto, Eduardo A. Fonseca

**Affiliations:** 1Hepatology and Liver Transplantation, Hospital Sírio-Libanês, São Paulo 01308-050, Brazil; aline_falleiros@hotmail.com (A.F.F.); renatasp59@gmail.com (R.P.S.P.); irene.miura@gmail.com (I.K.M.); veradanesi@icloud.com (V.L.B.D.); eliene_novais_oliveira@hotmail.com (E.N.O.); driporta@gmail.com (A.P.M.H.); cristianbvborges@gmail.com (C.B.V.B.); juvieiralobato@gmail.com (J.V.L.); gildaporta@gmail.com (G.P.); joaoseda@icloud.com (J.S.-N.); 2Hepatology and Liver Tranplantation, Santa Casa de Porto Alegre, Porto Alegre 90020-090, Brazil; flavia.feier@gmail.com

**Keywords:** pediatric liver transplantation, COVID-19, infection, outcomes, immunocompromised

## Abstract

Background: The COVID-19 infection has received the attention of the scientific community due to its respiratory manifestations and association with evolution to severe acute respiratory syndrome (SARS-CoV-2). There are few studies characterizing SARS-CoV-2 in pediatric immunocompromised patients, such as liver transplanted patients. The aim of this study was to analyze the outcomes of the largest cohort of pediatric liver transplant recipients (PLTR) from a single center in Brazil who were infected with COVID-19 during the pandemic. Methods: Cross-sectional study. Primary outcomes: COVID-19 severity. The Cox regression method was used to determine independent predictors associated with the outcomes. Patients were divided into two groups according to the severity of COVID-19 disease: moderate–severe COVID and asymptomatic–mild COVID. Results: Patients categorized as having moderate–severe COVID-19 were younger (12.6 months vs. 82.1 months, *p* = 0.03), had a higher prevalence of transplantation from a deceased donor (50% vs. 4.3%, *p* = 0.02), and had a higher prevalence of COVID infection within 6 months after liver transplantation (LT) (75% vs. 5.7%, *p* = 0.002). The independent predictor of COVID-19 severity identified in the multivariate analysis was COVID-19 infection <6 months after LT (HR = 0.001, 95% CI = 0.001–0.67, *p* = 0.03). Conclusion: The time interval of less than 6 months between COVID-19 infection and LT was the only predictor of disease severity in pediatric patients who underwent liver transplantation.

## 1. Introduction

Recently, the COVID-19 infection has received the attention of the scientific community due to its respiratory manifestations, association with evolution to severe acute respiratory syndrome (SARS-CoV-2), and high mortality [1]. To date [2], according to global reports, the number of cumulative confirmed cases of COVID-19 is 480,170,572, including 6,124,396 deaths. Brazil has long figured as the epicenter of the pandemic, registering the third largest number of confirmed cases and the second highest number of deaths. There are very few studies characterizing SARS-CoV-2 disease in the pediatric population, and those that have been published suggest that infected children (<18 years old) are less likely to progress to the severe form of the disease when compared to adults [3]. Even scarcer are morbidity and mortality data for immunocompromised children and adolescents, such as transplanted patients. During the pandemic, balancing the unknown risks of COVID-19 infection in the immunocompromised pediatric population and the known risk of high mortality in critically ill candidates on the waiting list (WL) represented a serious challenge for the continuity of transplant activity. In these circumstances, the severity status of the children on WL had to be taken into consideration when deciding to move forward with LT. For a brief period, we sailed uncharted waters. By constantly searching for COVID-free pathways, we were able to mitigate the risk of in-hospital contamination, allowing for safely continuing transplant activity [4]. However, the contamination of these patients in the community, in the post-transplantation period, could not be avoided. Knowledge of how the COVID-19 infection evolves in these immunocompromised pediatric patients plays an essential role in this scenario. The aim of this study was to analyze the clinical outcomes of the largest cohort of pediatric liver transplant recipients (PLTR) from a single center in Brazil who were contaminated with COVID-19 during the pandemic.

## 2. Patients and Methods

### 2.1. Pediatric Liver Transplantation Activity during the COVID-19 Pandemic

Activity during the pandemic was guided by standards that ensure safety for patients and donors in the case of living donor liver transplantation (LDLT), as detailed in our previous published study [4]. In short, during the pre- and immediate post-transplantation periods, after the establishment of a COVID-free pathway, patients with a higher risk of mortality on the WL were prioritized for LT. Epidemiological and clinical screening were performed, as well as real-time polymerase chain reaction for COVID-19 (PCR-RT for COVID-19) testing within 48 h of the planned LT. In LT with deceased donors, COVID screening was also required for both donor and recipient. At the beginning of the pandemic, we did not have access to rapid PCR-RT testing for the recipient, and in certain situations LT was performed even without negative diagnostic confirmation by COVID-19 testing when prolonged graft ischemia time risked rendering the organ unfeasible. These recipients with unknown COVID-19 status were kept in a holding isolation area and triaged into the COVID-free or COVID areas once the PCR-RT results came in. For LDLT, control measures against contamination by COVID-19 in the community were intensified for patients, donors, and caregivers prior to the LT. For hospitalized patients, access to visitors and family members was limited and restricted to only those with negative epidemiological and clinical screenings for COVID-19. The medical staff and healthcare workers involved in patient care who had epidemiological or clinical symptoms, and/or tested positive for COVID-19, were removed from contact with the patients. Telemedicine follow-up was prioritized for outpatients.

### 2.2. Study Design

Cross-sectional study in a single center for pediatric liver transplantation

### 2.3. Patient Selection and Recruitment

Between 1 March 2020 and 31 October 2021, clinical presentation and clinical course data from PLTR (≥0 and <18 years of age) were collected in person, or via telemedicine when the patient tested positive for COVID-19, presented symptoms suggestive of COVID-19 infection, had positive epidemiological screening (PLTR who had recent contact with symptomatic patients and/or contact with patients who tested positive for COVID-19), or presented positive in-laboratory screenings before the procedure (the incidental positive). COVID-19 was confirmed by PCR-RT for COVID-19 assay of nasal and pharyngeal swab specimens. No patient received COVID-19 vaccination as this was unavailable to age groups under 18 y.o. until the closing date of this study. The hospital’s ethics committees approved this study under protocol number HSL 2011-21.

Demographics and clinical characteristics included: gender, age, diagnosis, indication for LT, PELD (Pediatric End-Stage Liver Disease) score, LT type (living donor or deceased donor), time interval between LT and COVID-19 infection, and preexisting comorbidities (i.e., obesity, allergy, respiratory or cardiovascular disease, diabetes, malignancy, neurological disease, and delayed neuropsychomotor development). Presentation symptoms related to COVID-19 infection were grouped into asymptomatic, respiratory, or gastrointestinal symptoms; fever; headache; myalgia; anosmia and/or ageusia; multisystem inflammatory syndrome; and others.

### 2.4. Follow-Up Assessments

During the study period, the daily follow-up of hospitalized patients was carried out by accessing data on the clinical course related to organ failure, respiratory support (none, oxygen therapy with nasal catheter, high-flow, non-invasive ventilation, intubation), hemodynamic support with vasoactive medications, or need for renal replacement therapy. Clinical outcomes included survival and length of hospital and/or ICU stay. For outpatients, the periodic follow-up was aimed at the need and reason for hospitalization and outcomes. The severity of the COVID-19 disease was classified into 4 categories, according to a published study: [5] mild disease, including fever, cough, sore throat, runny nose and/or myalgia, with no dyspnea; moderate disease, including fever, dyspnea and/or chest imaging consistent with SARS-CoV-2 pneumonia, and no change from baseline respiratory support; severe disease, including all moderate disease markers plus increased need for supplemental oxygen support and/or ventilator support requirements; and critical disease, including respiratory failure requiring mechanical ventilation with high parameters, acute respiratory distress syndrome and/or shock, or systemic inflammatory response. Baseline immunosuppression data and their change were collected as necessary, as were data regarding the type of target therapies with a supposed effect on the modulation of COVID-19 infection, including azithromycin, heparin, hydroxychloroquine, and others.

### 2.5. Outcome Definitions

Primary outcomes: patient survival, disease severity categorized in the spectrum from moderate to severe, and need for respiratory support. Secondary outcomes: need for hospitalization for COVID-19 disease.

### 2.6. Statistical Analysis

Categorical variables were presented in numbers and percentages, and continuous variables were presented as means and standard deviations (SD), or as medians and interquartile ranges (IQR), as appropriate. Results were compared using *t*-tests or appropriate non-parametric tests when distributional assumptions were in doubt, and differences between groups were assessed using chi-square or Fisher’s exact tests, when needed. The Cox regression method was used to determine independent predictors associated with the outcomes categorized as moderate and severe COVID-19 disease. The variables that presented a *p*-value ≤ 0.10 in the univariate analysis, and those with clinical relevance, were included in the multivariate model. Significant differences were considered at a *p* < 0.05. All analyses were performed using the SPSS 21.0 statistical package (IBM Inc., Chicago, IL, USA)

## 3. Results

According to the selection criteria, 74 PLTR who tested positive for COVID-19 were recruited, with the following distribution: 42 (56.7%) who had symptoms related to COVID-19, 15 (20.3%) who had positive epidemiology for COVID-19, and 17 (23%) who had a positive test for COVID-19 at pre-procedural screening (incidental positive), with one of these prior to an LT with deceased donor.

A total of 74 PLTR with COVID-19 were included (Table 1 and Table 2); 38 (51.4%) were female, with a median age at COVID-19 infection of 81.2 (26.1–158.2) months, and median time interval between LT and COVID-19 infection of 60 months (IQR 14.7–84.5). Indications for LT included biliary atresia (BA) as the most frequent diagnosis in 48 (64.8%) patients, and living donor was the prevalent modality of LT. Regarding pre-existing comorbidities, there was a predominance of obesity, in 13 (17.5%) patients, followed by allergy, respiratory, and oncological diseases. Symptomatic patients totaled 49 (66.2%), and respiratory symptoms (cough and runny nose) were the most frequent presentation in 27 (36.4%), followed by gastrointestinal symptoms in 20 (36.4%), and fever in 19 (25.6%) patients. Until the end of the study, no patient underwent COVID-19 vaccination, according to the standards and recommendations of the National Vaccination Program in Brazil [6].

### 3.1. Clinical Course of Outpatients

Follow-up for 65 (87.8%) patients was done on an outpatient basis; most of them did not require changes in baseline immunosuppression, which was adjusted for only one patient, for whom mycophenolate was discontinued (Table 3). Target therapies with supposed effect against COVID-19 were administered in 17 (16.9%) patients, and azithromycin plus corticotherapy was the most frequent drug association. The hospitalization rate for COVID-19 treatment was 8.4%—in six patients due to respiratory distress or need for hydroelectrolytic replacement (or both, in two patients), and suspected sepsis or other causes (or both in, one patient). Of the remaining patients, 56 (86.1%) evolved uneventfully, 4 (6.1%) had persistent diarrhea (up to 30 days) and spontaneous resolution and 1 had multisystem inflammatory syndrome (MIS) 1 month after the diagnosis of COVID-19. No patient required hospitalization, and all had a good recovery.

### 3.2. Clinical Course of Hospitalized Patients

Analysis of the clinical courses of patients who were hospitalized showed a predominance of disease severity in its mild form in 55.6%—five patients (Table 4). Of the nine hospitalized patients in this cohort, three tested positive during hospitalization and one in the immediate post-LT period (deceased donor). This patient did not have a positive epidemiological or clinical screening for COVID-19; however, the PCR-RT result was not available until the moment of LT. This patient was extubated in the immediate post-LT period, remaining in a high-flow nasal cannula for 1 week. He required reintubation and remained on mechanical ventilation for 3 days, and then returned to respiratory support with the high-flow nasal cannula for another 9 days, as well as the use of vasoactive drugs for longer. This patient presented chest imaging consistent with SARS-CoV-2 pneumonia, classified in the severe category. No target therapy with supposed effect against COVID-19 nor change in immunosuppression was administered. Another patient tested positive in the immediate post-LT period—the12th day post-op—and required respiratory support with nasal oxygen therapy in the ICU, with the severity of the disease categorized as moderate.

Changing the basal immunosuppression of hospitalized patients was necessary in three (33.3%) patients, with mycophenolate discontinued in two and decrease in the level of tacrolimus in one patient (Table 4). Target therapies with supposed effect against COVID-19 were administered in four (44.4%) hospitalized patients, and the use of heparin was the prevalent approach. Renal replacement therapy and extracorporeal membrane oxygenation (ECMO) were not required in this cohort. The median length of hospital stay was 8 days (IQR 7–14) and for those who needed ICU support, it was 6 days (IQR 4–15).

### 3.3. Outcomes Analysis and Clinical Predictors of Severity of COVID-19

Classification of COVID-19 disease was moderate or severe in four (5.4%) patients (Table 1 and Table 2). These had lower weight at LT (*p* = 0.08), higher prevalence of DDLT (*p* = 0.02), were younger at COVID-19 diagnosis (*p* = 0.03), and had a lower median time interval between the LT and the diagnosis of COVID-19 (*p* = 0.03), especially in the analysis of an interval of <6 months (*p* = 0.002). There was no statistically significant correlation between the evolution of greater severity of the COVID-19 disease and the presence of comorbidities, presentation symptoms, or baseline immunosuppression such as the association of one, two, or three drugs, and the use of mycophenolate or steroids. There was no mortality in this cohort. The independent predictor of COVID-19 severity identified in the multivariate analysis, as shown in Table 5, was COVID-19 infection <6 months after LT (Hazard Ratio [HR] = 0.01; 95% CI = 0.001–0.67; *p* = 0.03). Age at COVID-19 infection and weight at LT were not statistically significant in the multivariate analysis.

## 4. Discussion

An established concept is that the presence of immunosuppression increases the risk of opportunistic infections as well as the severity of the clinical course of these infections. However, there are still doubts whether this perception can also be applied to the novel coronavirus disease. Initially, some published studies showed increased COVID-19 lethality in adults undergoing immunosuppression after solid organ transplantation [7,8], without determining the risk factor involved. The pediatric population, however, as recent studies show, seems to be less likely to progress to severe forms of SARS-CoV-2 infection [9,10], but for immunocompromised pediatric patients, especially children with severe liver disease, who have independent risk factors for mortality on the WL [11], we did not know which of the associations of SARS-CoV-2 infection in the early post-transplant period could potentiate the risk of mortality. The answer to such questions illuminates the paths to be followed, both in the maintenance of transplant activity and in the presumed outcomes of cases that acquired COVID-19 by community transmission. The results of this study show a high frequency of asymptomatic PLTR, similar to the immunocompetent pediatric population demonstrated in published studies [12,13]. The clinical presentation of symptomatic cases demonstrated the same influenza-like pattern of symptoms as the general pediatric population [13,14], and most patients were followed up and treated on an outpatient basis. The reason for this greater tolerance of children when compared to adults is still unknown, as well as whether immunosuppression alone could impair this specific resistance of the pediatric population against COVID-19 infection. There are few data available in the literature concerning COVID-19 infection in immunocompromised pediatric patients.

Recently, a study published on adult LT recipients showed a lower standardized mortality rate when compared to the general population [15], and the presence of comorbidities in these patients when compared to the general population could predispose to an expected severity of COVID-19 [16]. However, carrying this concept over to the immunosuppressed pediatric population does not seem logical since there are specificities in the comorbidities between populations. In adults, comorbidities at risk of COVID-19 severity are related to aging and prolonged exposure to immunosuppressants, such as in diabetes [17,18], chronic heart disease, arterial hypertension [19,20,21], and renal dysfunction [17,22,23]. The presence of comorbidities was found in almost half of the cohort studied, with obesity being the most prevalent, followed by allergy, respiratory diseases and, in equal proportion, oncological, cardiological, and neurological with NPMD dysfunctions, in this order. Obesity is recognized as a risk factor in both adults and children [24,25]. Children may have other comorbidities specific to their age group, such as asthma, congenital heart disease, and inborn errors of metabolism, that also determine a high risk for severity of COVID-19 [26,27]; however, none of the comorbidities found in this study were related to the severity of the outcome.

Regarding the clinical course, the outcomes of this cohort showed few severe cases and low hospitalization rate—of these, only two in the ICU—and absence of multiple organ failure. All patients showed recovery. The only predictor of severity of outcome was the time interval of less than 6 months between LT and COVID-19 infection. Three of the four recipients categorized in the moderate to severe spectrum of COVID-19 were infected within 6 months after LT. The plausible explanation is that in the first 6 months after LT the presence of greater immunosuppressive overload may explain a greater exposure to risk of disease severity due to increased viral load [28,29]. Another argument that would explain the unfavorable impact of early post-LT COVID-19 infection is that the consequences of chronic liver disease, such as malnutrition [29] and predisposition to secondary infections [30], as well as in the adult recipient’s renal dysfunction associated with liver disease [31], are still preponderant in the pediatric liver recipient and may aggravate the clinical evolution of COVID-19. Other studies have shown no correlation between outcomes and the LT interval and COVID-19 infection in adult [15] and pediatric [32,33,34] recipients. However, in such studies, patients diagnosed with COVID-19 within 6 months after LT were in small numbers.

Prolonged and stable maintenance of immunosuppressants for 6 months after LT, regardless of the combination (one, two, or three drugs) or immunosuppressant used (mycophenolate and/or steroids) in PLTR were not associated with worse outcomes. The proportion of mild and asymptomatic cases of COVID-19 in this cohort was mostly higher. So far, data from our study support the fact that immunosuppression, after the critical post-transplant period of 6 months appears not to be a risk factor for the worsening of COVID-19 in PLTR. If there is a beneficial effect of immunosuppression by modulation of the immune system, thus preventing progression to severe forms of the disease, as suggested by uncontrolled observational clinical studies [15,35,36,37,38,39,40], this should be confirmed soon with comparative and controlled studies. Some limitations of the present study should be noted. There was a lack of comparison between the data from the general pediatric population not submitted to immunosuppression. This was due to the unavailability, by government health agencies, of data with sufficient detail for the categorization of patient severity as well as outcomes. Such a comparative analysis could determine whether there is a higher risk of severity of COVID-19 infection in the immunocompromised pediatric population. Another limitation was the small proportion of unfavorable outcomes, specifically in the pediatric population, even in immunocompromised children, making it difficult to find predictors of the aggressiveness of COVID-19 disease. Multicentric, controlled clinical trials with large patient samples can bring new perspectives in this discussion.

## 5. Conclusions

PLTR receiving immunosuppression do not seem to have greater susceptibility to severe forms of COVID-19 when compared to the immunocompetent pediatric population. The time interval of less than 6 months between COVID-19 infection and LT was the only predictor of disease severity in pediatric patients.

## Figures and Tables

**Table 1 microorganisms-10-01030-t001:** Clinical characteristics of severity from 74 PLTR with COVID-19 infection. LT-related data.

Characteristic	Overall	Moderate + Severe COVID *n* = 4	Asymptomatic + Mild COVID *n* = 70	*p*
Age at LT, m, median (IQR)	12.7 (8.1 to 29.9)	12.4 (6.3 to 34.8)	12.7 (8.1 to 19.9)	0.61
Female, *n* (%)	38 (51.4)	1 (25)	37 (52.8)	0.33
Weight at LT, kilograms, median (IQR)	20.6 (13.4 to 41.2)	8.3 (5.9 to 48.8)	20.9 (13.7 to 41.2)	0.08
Indication for LT, *n* (%)				
biliary atresia	48 (64.8)			
other biliary diseases	9 (12.2)			
hepatoblastoma	4 (5.4)			
metabolic diseases	3 (4.1)			
acute liver failure	2 (2.7)			
autoimmune hepatitis	2 (2.7)			
other	6 (8.1)			
BA, *n* (%)	48 (64.8)	2 (50)	46 (65.7)	0.6
PELD score, median (IQR)	15 (6.5 to 19.5)	11.5 (6.5 to 42)	15 (6.5 to 19.5)	0.99
LT type, *n* (%)				
LDLT	69 (93.2)	2 (50)	67 (95.7)	0.02
DDLT	5 (6.7)	2 (50)	3 (4.3)	
Baseline Immunosuppression				0.47
None	1 (1.4)	0	1 (1.4)	
CNI	29 (39.2)	1 (25)	28 (40)	
CNI + My	32 (43.2)	2 (50)	30 (42.9)	
CNI + My + CS	12 (16.2)	1 (25)	11 (15.7)	
My use	25 (33.8)	1 (25)	24 (34.3)	1.0
CS use	32 (43.2)	3 (75)	29 (41.4)	0.31

Abbreviations—IQR: interquartile range; m: months; LDLT: living donor liver transplantation; DDLT: deceased donor liver transplantation; CNI: calcineurin inhibitor; My: mycophenolate; CS: corticosteroids.

**Table 2 microorganisms-10-01030-t002:** Clinical characteristics of severity from 74 PLTR with COVID-19 infection. LT and COVID data.

Characteristic	Overall	Moderate + Severe COVID *n* = 4	Asymptomatic + Mild COVID *n* = 70	*p*
Age at COVID, months, median (IQR)	81.2 (26.1 to 158.2)	12.6 (7.9 to 112.5)	82.1 (33.6 to 168.4)	0.03
LT and COVID: time interval (months), median (IQR)	46 (14.6 to 109.8)	1.16 (0.1 to 78.1)	46.7 (15.2 to 111.8)	0.03
LT and COVID: median time interval, *n* (%)				0.01
<6 m	7 (9.5)	3 (75)	4 (5.7)	
≥6–12 m	10 (13.5)	0	10 (14.3)	
≥12–60 m	30 (40.5)	0	30 (42.9)	
≥60–120	12 (16.2)	1 (25)	11 (15.7)	
>120 m	15 (20.3)	0	15 (21.4)	
LT to COVID < 6 months	7 (9.5)	3 (75)	4 (5.7)	0.002
≥6 months	67 (90.5)	1 (25)	66 (94.3)	
Comorbidities, *n* (%)				
none	40 (54)			
obesity	13 (17.5)	1 (25)	12 (17.1)	0.54
allergy	5 (6.7)	0	5 (7.1)	1.0
respiratory disease	4 (5.5)	0	4 (5.7)	1.0
oncological disease	3 (4.1)	0	3 (4.3)	1.0
cardiological disease	3 (4.1)	1 (25)	2 (2.9)	0.15
neurological disease + NPMD	3 (4.1)	0	3 (4.3)	1.0
other	8 (10.8)			
Symptoms (yes), *n* (%)	49 (66.2)	4 (100)	45 (64.3)	0.29
Symptoms presentation *n* (%)				
asymptomatic	25 (33.8)			
respiratory	27 (36.4)	3 (75)	24 (34.3)	0.13
gastrointestinal	20 (27)	1 (25)	19 (27.1)	1.0
fever	19 (25.6)	2 (50)	17 (24.3)	0.27
headache	8 (10.8)	1 (25)	7 (10)	0.37

Abbreviations—IQR: interquartile range; m: months; NPMD: neuropsychomotor developmental delay.

**Table 3 microorganisms-10-01030-t003:** Clinical course and outcomes of 65 outpatients PLTR with COVID-19 infection.

Characteristic	*n* (%)
Hospitalization rate and reason	6 (8.4%)
respiratory distress	2 (33.3%)
hydroeletrolitic replacement	2 (33.3%)
suspicion of sepsis	1 (16.7%)
other	1 (16.7%)
Baseline immunosuppression	
no change	64 (98.5%)
mycophenolate discontinuation	1 (1.5%)
Target therapies with supposed effect against COVID-19 ^a^	
none	54 (83.1%)
azithromycin	8 (12.3%)
corticosteroids	5 (7.7%)
heparin	1 (1.5%)
others	3 (4.6%)

Abbreviations—d: days. ^a^ most patients who underwent treatment used a drug combination.

**Table 4 microorganisms-10-01030-t004:** Clinical course and outcomes of 9 hospitalized PLTR with COVID-19 infection.

Characteristics	*n* (%)
Coming from the outpatient clinic	6 (66.7%)
Previously hospitalized	3 (33.3%)
for LT	2 (66.7%)
PTLD investigation	1 (33.3%)
Duration of symptoms, median days (IQR)	8 (6–8)
Severity of COVID-19 disease	
mild	5 (55.6%)
moderate	2 (22.2%)
severe	2 (22.2%)
critical	0
Respiratory support ^a^	
nasal oxygen therapy	2 (22.2%)
HFNC	1 (11.1%)
intubation/mechanical ventilation	1 (11.1%)
Hemodynamic/vasoactive support	1 (11.1%)
Baseline immunosuppression	
no change	6 (66.7%)
mycophenolate discontinuation	2 (22.2%)
decrease tacrolimus level	1 (11.1%)
Target therapies with supposed effect against COVID-19 ^b^	
none	4 (44.4%)
heparin	2 (22.2%)
azithromycin	1 (11.1%)
other	1 (11.1%)
Length of stay, median (IQR), days	
hospital	8 (7–14)
ICU	6 (4–15)
Outcome	
discharged	9 (100%)
died	0

Abbreviations—PTLD: post-transplant lymphoproliferative disorder; HFNC: high-flow nasal cannula; ICU: intensive care unit; IQR: interquartile range. ^b^ some patients who underwent treatment used a drug combination. ^a^ Some patients required more than one type of respiratory support.

**Table 5 microorganisms-10-01030-t005:** Multivariate Cox’s regression analysis for predictors of severe COVID-19 in PLTR.

Variables	HR	95% CI	*p*
COVID-19 infection <6 months after LT	0.01	(0.001 to 0.67)	0.03
Age at COVID-19 infection (months)	0.87	(0.68 to 1.12)	0.30
Weight at LT	1.4	(0.77 to 2.67)	0.24

Abbreviation—HR: hazard ratio.

## Data Availability

The data presented in this study are available on request from the corresponding author. The data are not publicly available due to patient confidentiality.

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
