# Peer review of "Impact of COVID-19 Infection on Children and Adolescents after Liver Transplantation in a Latin American Reference Center"

_microorganisms, 2022, doi:10.3390/microorganisms10051030_

Round 1
Reviewer 1 Report
Thank you for asking me to review this interesting paper, dealing with a really relevant topic and with interesting implications for children undergoing liver transplantation.
Overall, the manuscript is well-written and organized, with minor flaws and minor language issues (I suggest English revising). I believe the significance of content and interest for the reader are high.
I have some comments for the Authors:
Abstract:
- Line 16: contaminated > I think the term “infected” would be more appropriate, here and wherever used throughout the text.
Introduction:
Lines 33-34: add reference to the reported epidemiology.
The introduction section is otherwise adequate and complete.
Patients and Methods:
This section is well-constructed and developed. Especially, study design and outcome definitions are clearly stated.
Results:
Data are clearly reported, only a few marginal comments:
Line 133: What do you mean by positive epidemiology? Please rephrase.
Line 158: please, replace “performed” with “administered”. Please do not use the “perform a therapy” phrase throughout the text, please correct accordingly (eg. Lines 191-192).
Lines 162-164: Please rephrase the following sentence: “The remaining patients 56 (86.1%) evolved uneventfully, 4 (6.1%) had persistent diarrhea (up to 30 days) and spontaneous resolution and 1 had multisystem inflammatory syndrome (MIS) 1 month after the diagnosis of COVID-19” as follows: “Of the remaining patients, 56 (86.1%) evolved uneventfully, 4 (6.1%) had persistent diarrhea […]”
Line 175 and 187: HFNC:High-flow nasal canula > please correct with “cannula”.
Line 203: Renal replacement therapy and extracorporeal membrane oxygenation (ECMO) was not required in this cohort > please replace “was” with “were”.
Discussion:
This section is well-constructed, however limitations should be defined more clearly, and in a separate paragraph.
Moreover:
Line 233: a finding like that of the immunocompetent pediatric population> please, rephrase as follows: “similarly to the immunocompetent pediatric population”.
Lines 280-281: The sentence is very strong. I suggest rephrasing as follows: “So far, what we can say is that immunosuppression, after the critical post-transplant period of 6 months, is not a risk factor for the worsening of COVID-19 in PLTR.” > So far, data from our study support the fact that immunosuppression, after the critical post-transplant period of 6 months, appears not to be a risk factor for the worsening of COVID-19 in PLTR.
Conclusion:
Line 289: please rephrase: PLTR submitted to immunosuppression > PLTR receiving immunosuppression
Tables:
The tables are well-organized, informative and provide exhaustive data. However, to make it more readable, I would suggest splitting Table 1 it into two separated tables, one about LT-related data and one about LT and Covid data. Table 2 has typos (eg Chacteristic> Characteristic), please correct accordingly.

Reviewer 2 Report
Congratulations, this is an excellent paper, with sound methodology, interesting results and great presentation !
Author Response
Point 1: “Congratulations, this is an excellent paper, with sound methodology, interesting results and great presentation!”
Response 1: Thank you for your comments and the approval of the present manuscript.
Sincerely,
Eduardo Antunes da Fonseca, MD, PhD
Hepatology and Liver Transplantation - Hospital Sírio-Libanês
São Paulo, Brazil
Reviewer 3 Report
the manuscript covers an important covid-19 related argument still not well explored in literature.
moreover, the study enrolled a pediatric population, often not largely studied in the literature.
Covariates should be described both considering the methods used to collect as well as the methods used to operationalize them.
the setting should be described
ethical approval is not reported. please add
discussion is well written and interesting to read
could you add (in the discussion section) some public health considerations of your results
conclusions are coherent with the results.
